# The Influence of the LINC00961/SPAAR Locus Loss on Murine Development, Myocardial Dynamics, and Cardiac Response to Myocardial Infarction

**DOI:** 10.3390/ijms22020969

**Published:** 2021-01-19

**Authors:** Ana-Mishel Spiroski, Rachel Sanders, Marco Meloni, Ian R. McCracken, Adrian Thomson, Mairi Brittan, Gillian A. Gray, Andrew H. Baker

**Affiliations:** 1Centre for Cardiovascular Science, Queens Medical Research Institute, University of Edinburgh, Edinburgh EH16 4TJ, UK; am.spiroski@ed.ac.uk (A.-M.S.); rachel.sanders@ed.ac.uk (R.S.); marco.meloni@sanofi.com (M.M.); ian.mccracken@ed.ac.uk (I.R.M.); mbrittan@exseed.ed.ac.uk (M.B.); gillian.gray@ed.ac.uk (G.A.G.); 2Edinburgh Preclinical Imaging, Edinburgh Preclinical Imaging, BHF Centre for Cardiovascular Science, University of Edinburgh, Edinburgh EH16 4TJ, UK; adrian.thomson@ed.ac.uk

**Keywords:** lncRNA, LINC00961, SPAAR, scRNASeq, CRISPR/Cas9, cardiovascular physiology, fetal growth restriction, myocardial infarction

## Abstract

Long non-coding RNAs (lncRNAs) have structural and functional roles in development and disease. We have previously shown that the LINC00961/SPAAR (small regulatory polypeptide of amino acid response) locus regulates endothelial cell function, and that both the lncRNA and micropeptide counter-regulate angiogenesis. To assess human cardiac cell SPAAR expression, we mined a publicly available scRNSeq dataset and confirmed LINC00961 locus expression and hypoxic response in a murine endothelial cell line. We investigated post-natal growth and development, basal cardiac function, the cardiac functional response, and tissue-specific response to myocardial infarction. To investigate the influence of the LINC00961/SPAAR locus on longitudinal growth, cardiac function, and response to myocardial infarction, we used a novel CRISPR/Cas9 locus knockout mouse line. Data mining suggested that SPAAR is predominantly expressed in human cardiac endothelial cells and fibroblasts, while murine LINC00961 expression is hypoxia-responsive in mouse endothelial cells. LINC00961^–/–^ mice displayed a sex-specific delay in longitudinal growth and development, smaller left ventricular systolic and diastolic areas and volumes, and greater risk area following myocardial infarction compared with wildtype littermates. These data suggest the LINC00961/SPAAR locus contributes to cardiac endothelial cell and fibroblast function and hypoxic response, growth and development, and basal cardiovascular function in adulthood.

## 1. Introduction

Over 17 million deaths worldwide annually are due to cardiovascular diseases (CVD) [1]. With no clinically utilised therapeutics available to reverse the disease process, management focuses on reducing risk factors that exacerbate the “silent symptoms” which contribute to ischaemic diseases of the heart and vasculature. Myocardial infarction (MI), which causes a prolonged lack of blood flow to the heart muscle, ultimately resulting in tissue necrosis and formation of a fibrotic scar, is one such disease. Whilst an MI shortens a patient’s life expectancy by >16 years [2,3], greater acute survival rates increases the incidence of subsequent heart failure. In concert with higher worldwide life expectancies and burgeoning increases in CVD risk factors, the economic and social burden of disease is ever-increasing. Investigating the underlying mechanisms which contribute to cardiovascular dysfunction and identifying potential therapeutic targets provides the most impactful opportunity for reducing the compounding costs of CVD.

As approximately 98% of the human genome that is transcribed into RNA does not code for protein [4], non-coding RNAs (ncRNA) provide a possible target to investigate both their physiological relevance and therapeutic capacity as novel interventional approaches. Long non-coding RNAs (lncRNAs) can be within a known protein-coding gene (intronic) and over-lapping with another gene (sense-overlapping), antisense to a gene they are likely to regulate (present on the opposite DNA strand, sequence overlapping), or bi-directional to a gene they likely regulate (present on the opposite DNA strand, sequences not over-lapping). LncRNAs located between neighbouring protein-coding genes are termed long intergenic non-coding RNA (lincRNA). Various lincRNAs have been reported to contribute to cardiovascular biology, with their functions contributing to embryological heart development, cardiovascular cell commitment, cell migration, smooth muscle cell (SMC) phenotype-switching, vascular endothelial cell (EC) commitment, and angiogenesis [5,6,7,8,9,10].

LINC00961, a lincRNA located on human chromosome 9 at p13.3, has 4 predicted transcripts with 1 predicted to undergo non-sense-mediated decay (ENSG00000235387). The locus is 1557 base pairs (bp) and contains 2 exons, of which exon 2 encodes the 75 amino acid micropeptide previously termed small regulatory polypeptide of amino acid response (SPAAR) by Matsumoto and colleagues, who identified 2 potential transcription start sites [11]. The mouse homologue (previously 5430416O09Rik) maps to chromosome 4 with 65.33% amino acid sequence conservation. Although greater conservation is observed between humans and non-human primates, the mouse and human locus is conserved in synteny with many of the same neighbouring protein-coding genes. Interestingly, Di Salvo and colleagues show a 1.4 log2 (fold change) in the right ventricle of heart failure patients compared to healthy donor controls [12]. Additionally, LINC00961 is expressed in highly vascular tissues such as the kidney, lungs, and placenta, and SPAAR deletion results in a 15.3-fold increase in LINC00961 locus expression in the murine heart [11]. Thus, based on the association of LINC00961 with vascular endowment, the loss of LINC00961 could influence structure and function from prenatal life onward. From reduced vascular enrichment in the placenta and its influence on foetal and postnatal growth and development, to adult disease risk and pathological response, particularly in the cardiovascular and pulmonary systems, LINC00961 loss of function could have clinical implications throughout the lifespan.

LINC00961 has been identified in multiple disease pathways associated with cardiovascular dysfunction. Wu and colleagues recently reported that LINC00961 is downregulated in patients with coronary artery disease (CAD) and in APOE^–/–^ mice [13]. In response to hypoxia, the H9C2 rat cardiomyoblast cell line demonstrates increased signal transducer and activator of transcription 1 (STAT1) interaction with the LINC00961 promoter and subsequent LINC00961 expression [14]. Downstream phosphorylation of phosphoinositide 3-kinase (PI3K), protein kinase B (AKT), and glycogen synthase kinase-3β (GSK3β) were also inhibited. Activation of this pathway has been implicated in abrogating the pathological remodelling of the heart post-hypoxia/reperfusion injury [15], suggesting LINC00961′s potential contribution to cardiomyocyte maintenance. Additionally, we have previously demonstrated that the LINC00961 locus directly regulates endothelial cell (EC) function, with SPAAR and LINC00961 counter-regulating angiogenesis in vitro and evidenced in a murine hindlimb ischaemia model of critical limb ischaemia [16]. In our previous work, we showed that LINC00961 binds the actin-binding protein thymosin beta-4x, which is involved in regulating cytoskeletal dynamics, angiogenesis, and cell migration [17], while SPAAR binds the actin-binding protein spectrin repeat-containing nuclear envelope protein 1 (SYNE1/NESPRIN-1), which is highly expressed in cardiomyocytes [18]. These initial data suggest the importance of the LINC00961/SPAAR locus in cellular maintenance and response to environmental insult. However, the cardiac effects of LINC00961 locus knockout (KO) in vivo are still unknown. Therefore, we investigated the expression of SPAAR with single-cell RNA sequencing (scRNAseq) data mining of human cardiac cell populations, the expression of the LINC00961 locus in a mouse endothelial-like cell line, and the effects of CRISPR/Cas9 LINC00961 locus deletion on murine growth and development, basal cardiac function, risk area following acute cardiac ischaemia, and physiological function following 7 days of chronic cardiac ischaemia in a surgical model of MI.

## 2. Results

### 2.1. Human Cardiac SPAAR Expression and the Endothelial Cell LINC00961 Response to Hypoxia

Mining of the Human Cell Atlas global heart dataset revealed SPAAR expression primarily in human cardiac EC (Figure 1A,B). LINC00961 expression is significantly reduced in murine bEnd.3 cells following 24 and 48 h hypoxia (1% O_2_) exposure compared with cells cultured in normoxia (5% O_2_) (Figure 1C,D).

### 2.2. Post-Weaning Growth and Development

Whilst there was no effect of genotype on female weight gain post-weaning (Figure 2A), longitudinal weight gain was delayed in male LINC00961^–/–^ mice (Figure 2B). There was an effect of sex and genotype on adult body weight, and a sex × genotype interaction, with male LINC00961^–/–^ mice attaining a lower adult weight (Table 1). Male LINC00961^–/–^ mice also had a greater brain to body weight ratio in adulthood, with a sex × genotype interaction. LINC00961^–/–^ mice had reduced tibia and head lengths, and absolute liver weight, with no difference in relative liver weight.

### 2.3. Basal Cardiac Dimensions and Function

Measures of cardiac function with ultrasound echocardiography left ventricular mitral/aortic blood flow, stroke volume (SV), ejection fraction (EF), fractional area change (FAC), and cardiac output (CO) were not different in WT and LINC00961^–/–^ mice (Figure 3A–D) at baseline. Adult LINC00961^–/–^ mice had significantly smaller left ventricular (LV) systolic and diastolic area, and lower end-systolic volume (ESV) and end-diastolic volume (EDV) (Figure 3E–H). Mitral valve function was not different in LINC00961^–/–^ compared with WT.

### 2.4. Acute Myocardial Risk Area

There was a significant increase in risk area 30 min after left anterior descending (LAD) coronary artery ligation (~17%, *p* < 0.01) in adult male LINC00961^–/–^ compared with WT mice (Figure 4A,B).

### 2.5. Cardiac Function Following Surgically Induced Myocardial Infarction

There was no difference between WT and LINC00961^–/–^ mice in measures of cardiac function by ultrasound echocardiography with Doppler flow (SV, EF, FAC, CO, ESV, or EDV) at 7 days post-MI (Figure 5A–F). Mitral valve function was not different in LINC00961^–/–^ compared with WT mice (data not shown).

## 3. Discussion and Conclusions

### 3.1. Discussion

We have shown that genetic LINC00961^–/–^ reduces adult LV area, ESV, and EDV, suggesting an overall reduction in LV size, and greater risk area following MI. We have previously reported that LINC00961 locus CRISPR/Cas9 KO reduces alpha-smooth muscle actin (αSMA) cell-density in the mouse hindlimb skeletal muscle without altering capillary density [16]. Global dysfunction in EC endowment resulting in compromised development of the vascular network could limit the capacity for myocardial growth and the maintenance of functional myocardium throughout life [20]. Reduced LV volume independent of adult size in LINC00961^–/–^ mice suggests that the vascular influence on cardiac development requires further investigation. Regardless, an appropriate relative heart size and adequate basal cardiac function suggests that at this relatively young age, the heart is able to adapt to reduced LV volumes and maintain physiological homeostasis.

The novel finding of asymmetric, delayed postnatal growth and a brain-sparing phenotype in male LINC00961^–/–^ offspring indicates that the reduced vascular endowment previously reported in adults likely compromises embryonic implantation, the development of foetal vasculature, and subsequent growth [21,22]. This foetal growth restriction (FGR) phenotype is the result of preferential shunting of blood away from peripheral organs in order to protect brain growth and development in utero. As inadequate endothelial endowment results in chronic foetal ischaemia and foetal oxygenation, the foetal adaptation to an inhospitable in utero environment results in the pathological programming of cardiovascular function [23]. Although females are relatively protected from environmental insults on the developing embryo [21], adult manifestations of hypoxic pregnancy include impaired ventricular relaxation, enhanced myocardial contractility, and cardiac sympathetic dominance, which contribute to maintained cardiac output [23,24,25,26,27,28]. Overall, these data suggest a role for the LINC00961/SPAAR locus in somatic growth and development, and adult cardiovascular function.

Establishment of an efficient microvascular network in order to supply blood flow matched to metabolic demand is necessary throughout the body; however, nowhere more so than in the continuously active myocardium. We showed that human cardiac endothelial cells and fibroblasts express SPAAR, and that LINC00961 expression in murine endothelial-like cells is hypoxia-responsive. These data suggest the LINC00961/SPAAR locus is expressed in basal conditions, and that regulation of the LINC00961 transcript is evident in response to environmental insults, such as those associated with pro-angiogenic states. We have previously reported that the LINC00961/SPAAR locus contributes to angiogenesis in the peripheral musculature, and here we have shown the potential for its influence on in vivo cardiovascular function and response to localised cardiac ischaemia. As we focused on the acute cardiac remodelling window, progression toward heart failure was outside of the scope of this current study. The apparent “normalisation” of left ventricular volume and area in LINC00961^–/–^ compared with wildtype mice at 7 days post-MI may suggest a relative acceleration in ventricular dilation in KO mice. Although LINC00961^–/–^ structural and functional cardiac deficiencies do not compromise homeostatic regulation at this young age, we hypothesise that disease severity and progression toward heart failure may be more pronounced during pathological remodelling. Future work investigating LINC00961 over-expression, for example using viral vector-based approaches for delivery to the heart, would elucidate the suitability of this lncRNA as a potential therapy.

Microvascular dysfunction is associated with cardiovascular dysfunction, accelerated pathological processes, and increased mortality [29,30]. ECs contribute to the regulation of shear stress, permeable barrier maintenance, leukocyte extravasation, blood clotting, inflammation, vascular tone, extracellular matrix (ECM) deposition, and vasoconstriction and vasodilation [31,32,33,34]. By adapting to altered blood flow and composition, EC activation primes the tissue environment for coagulation, inflammation, and vasoconstriction, a necessary cascade for wound healing and repair [31]. Inadequate vascular endowment has the potential not only to influence cardiac function, but also the tissue response to pathological processes. Future work would benefit from assessment of sex-specific assessment of initial cardiac risk area, either directly or by proxy with a troponin assay. This study is limited by the CRISPR approach to global LINC00961 KO, rather than with a conditional approach. Due to the influence of LINC00961 on vascular endowment and subsequent somatic growth and development, future work would benefit from a conditional KO, either inducible or at the organ-specific level.

### 3.2. Conclusions

We have shown that the left ventricular risk area following an acute MI is greater in LINC00961^–/–^ mice compared to wildtype littermates. However, at 7 days post-LAD ligation in an MI model, cardiac function in LINC00961^–/–^ was not different from wildtype littermates. These data suggest that the cardiac vascular network may be compromised, but the progression toward heart failure, with a significant ~40% reduction in ejection fraction, is not compounded by LINC00961^–/–^. It has previously been reported that reduced coronary vascularity increases cardiac vulnerability to fibrosis [35,36,37]. Reduced blood flow to the functional myocardium compromises coronary flow reserve required during increased cardiovascular work and pathological conditions [30,38]. Future investigations on the microvascular network during the acute remodelling window post-MI will help to identify the contribution of LINC00961 and SPAAR to extracellular matrix remodelling and the secretion of matrix metalloproteinases, fibronectin, and proteoglycans during cardiac scar maturation. As SPAAR expression has already been detected in the human heart [39], it will be important to understand the effect of the peptide, specifically on cardiac function, and whether this is counterbalanced by the LINC00961 transcript.

## 4. Materials and Methods

### 4.1. Ethical Approval

All animal procedures were performed at a University of Edinburgh Biomedical and Veterinary Sciences facility. This research was conducted in accordance with the Animals (Scientific Procedures) Act 1986 Amendment Regulations 2012, following ethical review by the University of Edinburgh Animal Welfare and Ethical Review Board (AWERB), under project (70/8933) and personal licenses held within the University of Edinburgh and conducted in accordance with ARRIVE (Animal Research: Reporting of In Vivo Experiments) guidelines [40]. To align with the National Centre for the Replacement, Refinement, and Reduction of Animals in Research (NC3Rs) principles, both male and female mice were used for this work.

### 4.2. Single Cell RNASeq Data Mining

Normalised SPAAR expression uniform manifold approximation and projection for dimension reduction (UMAP) visualisation was extracted from the Heart Cell Atlas global heart dataset (www.heartcellatlas.org), a publicly available database maintained within the scope of the Human Cell Atlas project [19] of scRNASeq cardiac cell data.

### 4.3. In Vitro Endothelial Cell Response to Hypoxia

Murine brain microvascular endothelial-like cell line bEnd.3 (ATCC) cells were cultured in Dulbecco’s Modified Eagle Medium (DMEM; Life Technologies, Glasgow, UK) with 10% FBS, and 1% penicillin and streptomycin at 37 °C and 5% CO_2_. Cells were plated in 6-well plates and incubated overnight to allow to adhere. The media was replaced the following day, and cells were incubated in a hypoxic chamber (Coy Laboratory Products, Grass Lake, Michigan, USA) at 1% O_2_ (5% CO_2_ and 94% N_2_) for 24 or 48 h. Three biological replicates were collected for each timepoint and condition. Cells were lysed with QIAzol (QIAGEN, Crawley, UK), RNA was extracted, and qRT-PCR was run, as previously described [6,16].

### 4.4. Generation of Experimental Groups

The LINC00961 locus knockout (KO) mouse line was generated using CRISPR/Cas9 technology (gRNA, proximal: ATACACTCCTCGCTCAATGT; gRNA, distal: CGAGGCTACGCTGTCAGTACT) on a C57BL/6nTAC genetic background by Taconic Biosciences (USA). KO and wild-type (WT) littermates were produced through heterozygous breeding, offspring ear clip samples were collected and genotyped by Transnetyx (Cordova, Tennessee, USA), as previously described [16], and experimentation was conducted as follows (Figure 6). A subset of females (WT, *n* = 2–12; KO, *n* = 8–13) and males (B: WT, *n* = 6–11; KO, *n* = 2–7) were weighed weekly, from 3–9 weeks of age, culled by cervical dislocation, and tissues were harvested at 9 weeks (female: WT and KO, both *n* = 9; male: WT, *n* = 10, KO *n* = 9) to analyse longitudinal growth and adult organ size. Adult female mice were allocated to undergo cardiac ultrasound echocardiography at 8 weeks of age to assess basal cardiac function (WT and KO, both *n* = 14), and a subset of these underwent surgical coronary artery ligation (CAL) to assess the in vivo physiological response to a heart failure procedure (WT and KO, both *n* = 6). Adult male mice were allocated to undergo CAL (WT and KO, both *n* = 6) to assess the in vivo cardiac response to acute cardiac ischaemia. Tissues from these offspring were utilised in other studies.

### 4.5. Cardiac Ultrasound Echocardiography

Basal cardiac function was acquired by ultrasound echocardiography with Doppler flow under isoflurane anaesthesia (4% induction, ~1.75% maintenance) on the Vevo 770 and Vevo 3100 preclinical imaging systems and analysed in Vevo 770 V3.0 and Vevo lab V3.2.6 image analysis software (FUJIFILM VisualSonics, Inc., Toronto, Canada) following independent in-house confirmation of individual system analytic outcomes (data not shown). Post-CAL cardiac function (7 days) was acquired on a Vevo 3100. Left ventricle (LV) function was assessed with brightness mode (B-mode), Pulse Wave Doppler (PWD), and motion mode (M-mode) in parasternal long axes, short axes and apical 4 chamber view (as appropriate), and EKV, ECG-gated Kilohertz Visualisation. All echocardiography analyses were blinded.

### 4.6. Acute Coronary Artery Ligation

Male mice at 8 weeks of age were anaesthetised with 70 mg·kg^−1^ intraperitoneal pentobarbital sodium (Euthatal), intubated endotracheally, and ventilated with 3 cm H_2_O positive-end expiratory pressure, with ventilation ventilated at 110 breaths per minute (tidal volume dependent on weight, 125–150 µL). Mice were maintained on homeothermic heading pads (Physitemp, Clifton, NJ, USA) and depth of anaesthesia was monitored with corneal and withdrawal reflexes. The CAL was conducted as previously described [41]. Briefly, the heart was exposed via left sternal thoracotomy, the epicardium was punctured to reveal the main branch of the LAD coronary artery, and the LAD occluded by looping a 7-0 prolene suture. Following 30 min ischaemia, the aorta was catheterised, the LAD suture loosened, and the mouse perfused with PBS. Following PBS perfusion, Evans blue dye was perfused, hearts were collected, processed as previously described, and stained with 2% triphenyltetrazolium chloride (TTC) to assess the cardiac risk area and infarct area [42]. Briefly, stained hearts were cut into 1 mm transverse sections to the level above the suture, incubated in 2% TTC for 30 min at 37 °C, blotted dry, and post-fixed in 4% formalin. Sections were scanned, and risk and infarct area were quantified manually in Fiji by an experienced user (Dr Spiroski).

### 4.7. Chronic Coronary Artery Ligation

Female mice at 8 weeks of age were anaesthetised with intraperitoneal 100 mg·kg^−1^ ketamine (Velatar, Boehringer Ingelheim, Berkshire, UK) + 10 mg·kg^−1^ xylazine (Rompun. 2%, Bayer, Berkshire, UK), intubated endotracheally, ventilated with 3 cm H2O positive-end expiratory pressure, and ventilated at 110 breaths per minute (tidal volume dependent on weight, 125–150 µL). Eye lubricant (Lacri-Lube Eye Ointment 5g, Allergan, Marlow, UK) and mice were maintained on homeothermic heading pads (Physitemp, Clifton, NJ, USA) and depth of anaesthesia was monitored with corneal and withdrawal reflexes. To induce chronic CAL, the heart was exposed via a left thoracotomy between the second and third ribs performed by blunt dissection, the epicardium was punctured to reveal the main branch of the LAD coronary artery, and the LAD was permanently occluded with a 7-0 prolene suture (Henry Schein, Gillingham, UK). After thoracic and skin closure with 6-0 prolene sutures (Henry Schein, Gillingham, UK), the anaesthetic was reversed with 1.0 mg·kg^−1^ subcutaneous atipamezole (Antisedan, Henry Schein, Gillingham, UK) and 500 uL 0.9% sterile saline. Mice were provided with 0.20 mg·kg^−1^ buprenorphine (Temgesic, Henry Schein, Gillingham, UK) at the time of surgery, 24 and 48 h post-operative, and housed in individually ventilated cages in a heat cabinet for 24 h to help maintain body temperature.

### 4.8. Statistical Analysis

Data analyses were blinded. Power calculations based on previous datasets were performed to determine the minimum sample size required to achieve statistical significance. For CAL procedures, with a power of 80% and a 5% chance of Type I error, 6 successful animals/group are needed to achieve significant differences in LV function in the chronic MI model. At an 80% survival rate, 8 animals/group are needed. With echocardiography as a clinically relevant endpoint for this study, with a power of 80% and a 5% chance of Type I error, 12 successful animals/group are needed to achieve significant differences in LV function for basal echocardiography procedures.

Data were analysed in JMP 12 (SAS Institute, Inc., Cary, North Carolina, USA). Distribution was verified with the Shapiro–Wilk test and non-parametric data were log-transformed where necessary. Sex comparisons and sex by genotype interactions were analysed by factorial analysis of variance (two-way ANOVA). Tukey’s post hoc testing was conducted where appropriate. Data are presented as the mean ± SEM.

## Figures and Tables

**Figure 1 ijms-22-00969-f001:**
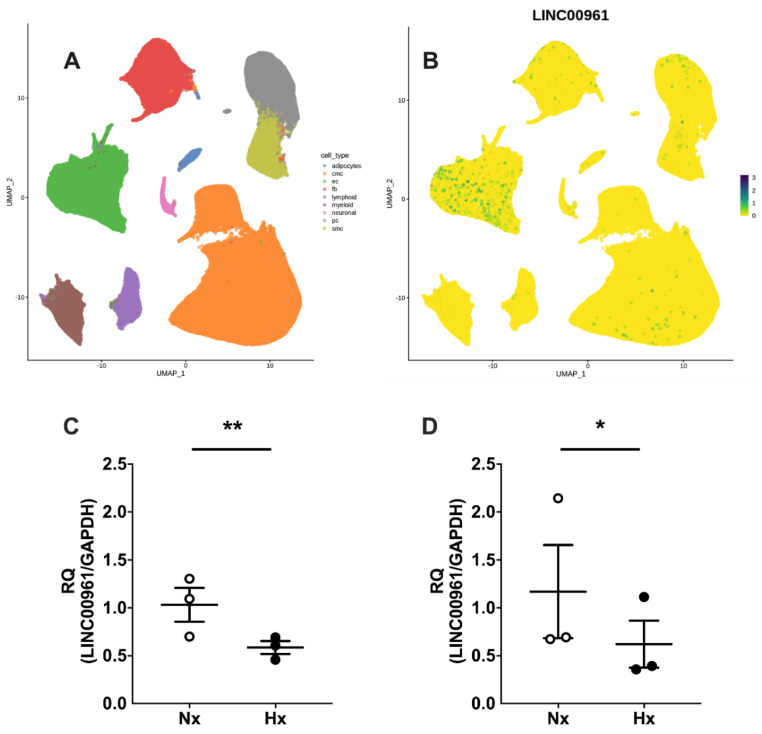
Human cardiac SPAAR (small regulatory polypeptide of amino acid response) expression and LINC00961 response to hypoxia. Uniform manifold approximation and projection for dimension reduction (UMAP) visualisation of (**A**) human cardiac endothelial cells (EC, green), fibroblast (FB, red), pericyte (PC, grey), SMC (olive), cardiomyocyte (CMC, orange), adipocyte (blue), neuronal cell (pink), myeloid (brown), and lymphoid (purple) fraction [19], and (**B**) SPAAR expression in human cardiac cell populations. LINC00961 expression is reduced in bEnd.3 mouse cells following 24 h (**C**, black *n* = 3) and 48 h (**D**, black *n* = 3) exposure to hypoxia (H, 1% O_2_) compared with bEnd.3 cells cultured in normoxia (Nx, 5% O_2_) collected in parallel (both *n* = 3). Student’s *t*-test. Data are mean ± SEM. * *p* < 0.05, ** *p* < 0.01.

**Figure 2 ijms-22-00969-f002:**
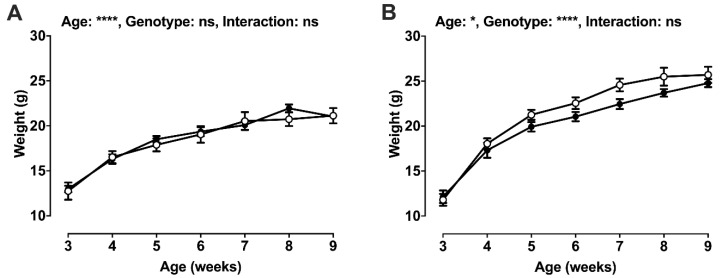
Post-weaning growth by sex. Weight gain from weaning (3 weeks) to 9 weeks of age in females (**A**: WT, white, *n* = 2–12; KO, black, *n* = 8–13) and males (**B**: WT, white, *n* = 6–11; KO, black, *n* = 2–7); 2-way ANOVA. Mean ± SEM. * *p* < 0.05, **** *p* < 0.0001, ns = not significant.

**Figure 3 ijms-22-00969-f003:**
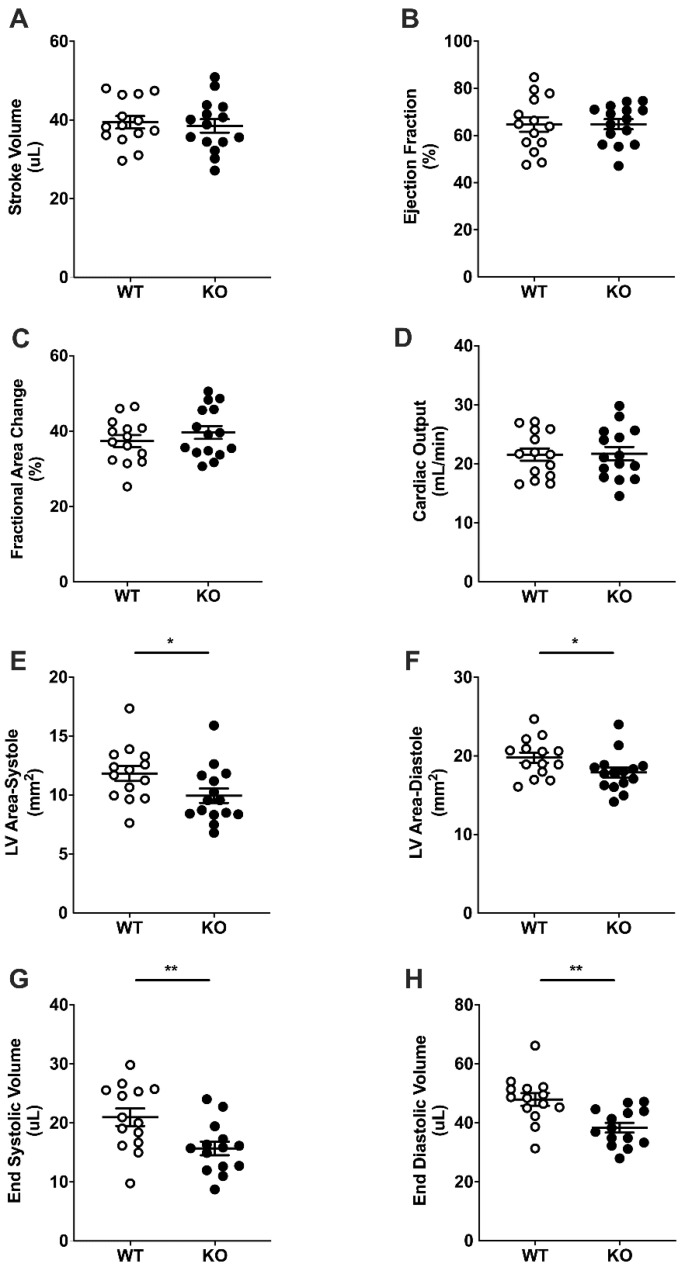
Basal cardiac dimensions and function by ultrasound echocardiography. Stroke volume (SV, **A**), ejection fraction (EF, **B**), fractional area change (FAC, **C**), cardiac output (CO, **D**), left ventricular systolic (**E**) and diastolic (**F**) areas, end systolic volume (ESV, **G**) and end diastolic volume (EDV, **H**) in 8-week-old female WT (white, *n* = 14) and KO (black, *n* = 14) mice. Student’s *t*-test. Data are mean ± SEM. * *p* < 0.05, ** *p* < 0.01.

**Figure 4 ijms-22-00969-f004:**
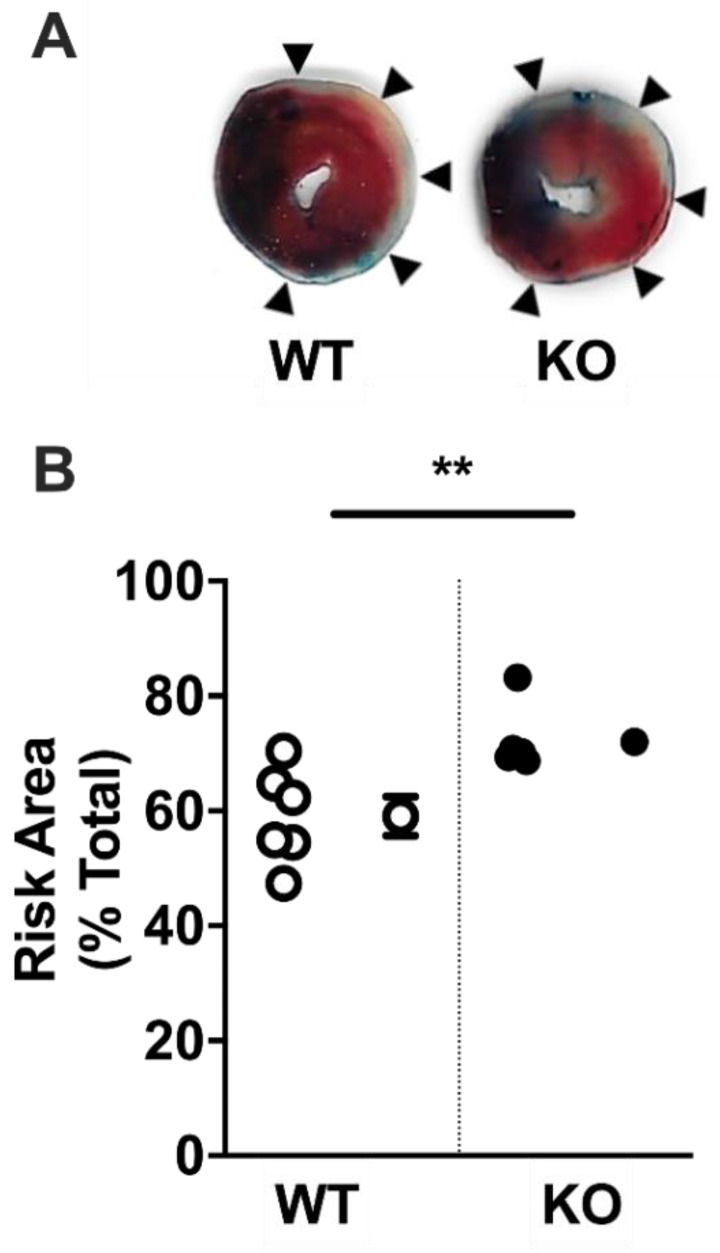
LINC00961 knockdown increases area at risk of myocardial infarction. (**A**) Myocardial infarction risk area (pink and white, indicated by arrows) relative to total heart area (blue, pink, and white) in heart tissue collected 30 min after left anterior descending coronary artery ligation in 9-week-old male mice. (**B**) Individual (left) and mean areas at risk in WT (white) and KO (black) mice (both *n* = 6). Student’s *t*-test. Data are mean ± SEM. ** *p* < 0.01.

**Figure 5 ijms-22-00969-f005:**
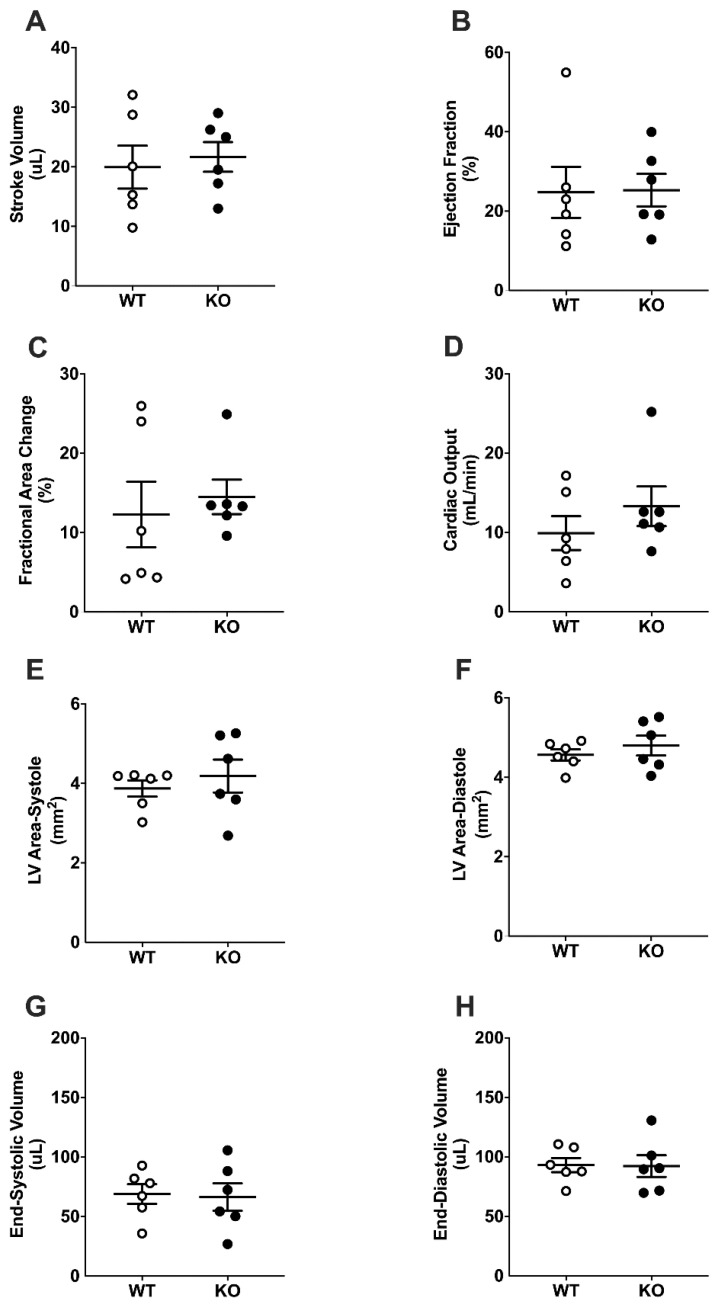
Cardiac dimensions and function following myocardial infarction (MI). SV (**A**), EF (**B**), FAC (**C**), CO (**D**), left ventricular systolic (**E**) and diastolic (**F**) areas, ESV (**G**) and EDV (**H**) in WT (white, *n* = 6) and KO (black, *n* = 6) female mice 7 days and 14 days following chronic LAD ligation. Student’s *t*-test. Data are mean ± SEM.

**Figure 6 ijms-22-00969-f006:**
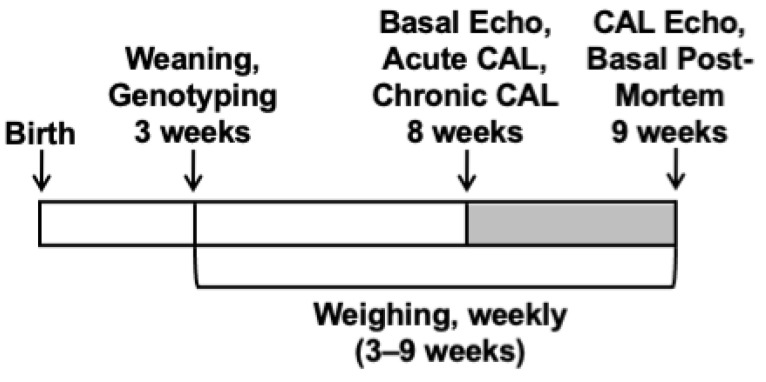
Schematic of age at experimental time points: Weaning and genotyping (3 weeks); basal ultrasound echocardiography (Echo) with Doppler flow, acute coronary artery ligation (CAL), chronic CAL induction (all 8 weeks); CAL Echo, and basal post-mortem (9 weeks); weekly weighing (3–9 weeks).

**Table 1 ijms-22-00969-t001:** Adult post-mortem organ weights.

	Female		Male		Significance (*p*)		
	WT *n* = 9	KO *n* = 9	WT *n* = 10	KO *n* = 9	Sex	Genotype	Interaction
Weight (g)	20.8 ± 0.9	20.9 ± 0.5	27.1 ± 0.6	23.7 ± 0.9	0.01	0.03	0.02
Brain (mg)	443.9 ± 8.8	428.4 ± 10.6	450.7 ± 5.3	435.5 ± 12.5	ns	ns	ns
Brain:Body weight (mg/g)	21.5 ± 0.5	20.5 ± 0.5	16.7 ± 0.4	18.5 ± 0.5	0.01	ns	0.009
Heart (mg)	124.2 ± 7.2	125.7 ± 9.6	137.0 ± 6.3	115.2 ± 6.6	ns	ns	ns
Heart:Body weight (mg/g)	6.0 ± 0.4	6.1 ± 0.4	5.0 ± 0.3	4.9 ± 0.3	0.03	ns	ns
Lungs, total (mg)	167.7 ± 16.5	147.1 ± 7.4	148.0 ± 3.6	139.9 ± 4.9	ns	ns	ns
Lungs:Body weight (mg/g)	7.5 ± 0.3	7.1 ± 0.3	5.5 ± 0.2	5.9 ± 0.3	0.01	ns	ns
Liver (mg)	1212 ± 64	1104 ± 34	1111 ± 71	1174 ± 69	ns	0.004	ns
Liver:Body weight (mg/g)	58.4 ± 2.5	53.2 ± 2.5	53.5 ± 2.4	49.6 ± 2.5	ns	ns	ns
Spleen, total (mg)	88.3 ± 5.8	88.0 ± 3.8	84.9 ± 4.3	80.3 ± 8.6	ns	ns	ns
Spleen:Body weight (mg/g)	4.2 ± 0.2	4.2 ± 0.2	3.1 ± 0.2	3.4 ± 0.2	0.009	ns	ns
Soleus, total (mg)	6.4 ± 0.4	6.4 ± 0.6	8.6 ± 0.6	8.1 ± 0.8	ns	ns	ns
Soleus:Body weight (mg/g)	0.31 ± 0.02	0.31 ± 0.03	0.32 ± 0.02	0.34 ± 0.03	ns	ns	ns
	**WT** ***n* = 5**	**KO** ***n* = 8**	**WT** ***n* = 10**	**KO** ***n* = 8**	**Sex**	**Genotype**	**Interaction**
Nose to anus (mm)	79.5 ± 1.5	79.7 ± 1.1	82.3 ± 1.1	79.0 ± 0.8	ns	ns	ns
Head length (mm)	25.0 ± 0.6	26.7 ± 0.3	26.4 ± 0.2	26.5 ± 0.6	ns	0.006	ns
Biparietal diameter (mm)	11.6 ± 0.2	11.2 ± 0.2	11.8 ± 0.1	11.7 ± 0.2	ns	ns	ns
Tibia length (mm)	15.4 ± 1.1	18.4 ± 0.7	17.5 ± 0.5	17.0 ± 0.4	ns	0.05	ns

Post-mortem organ weights were collected in female (WT, *n* = 5–9; KO, *n* = 8–9), and male (WT, *n* = 10; KO, *n* = 8–9) adults (9.7 ± 0.3 and 9.9 ± 0.3 weeks, respectively). Data are mean ± SEM. Statistical differences are indicated: main effect of sex, main effect of genotype, and sex × genotype interaction (two-way ANOVA with Tukey’s post-hoc comparison).

## Data Availability

The data presented in this study are available on request from the corresponding author. The data within the Heart Cell Atlas project are publicly available.

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
