# Peer review of "The Influence of the LINC00961/SPAAR Locus Loss on Murine Development, Myocardial Dynamics, and Cardiac Response to Myocardial Infarction"

_ijms, 2021, doi:10.3390/ijms22020969_

Round 1

Reviewer 1 Report

Authors present relevant preclinical data with respect to LINC00961/SPAAR locus loss on myocardial dynamics and cardiac response to myocardial infarction. They have previously identified that SPAAR is predominantly expressed in human cardiac endothelial cells and fibroblasts, while murine LINC00961 expression is hypoxia-responsive in mouse endothelial cells. Therefore, both locuses might have relevant implications for the development of cardiovascular disease. Their main result is that LINC00961-/- knockout mice showed a sex-specific delay in longitudinal growth and development, smaller left ventricular end- systolic and -diastolic areas and volumes, and greater risk area following myocardial infarction compared with wildtype littermates. Altogether, these data suggest a role for the LINC00961/SPAAR locus in cardiac endothelial cell and fibroblast cell function and hypoxic-response, as well as their development.

Ethical disclosures and animal protection protocols have been declared in the manscript and are well-described.

Methods are sufficiently described with no particular objections on my behalf.

I only have some minor comments:

  1. What would be possible clinical implications of your findings? I think it would be worthwhile to discuss possible outlook of your findings with respect to potential pharmacotherapeutics/treatment approaches in the future.
  2. Were there any limitations to your study?
  3. How was myocardial area-at-risk estimated? Who did these observations and was there any inter-observer or intra-observer reliability issues? This should be explained at greater details.
  4. In clinical practice, MIs are usually accompanied with an increase in end-systolic and end-diastolic volumes and pressures? This clinical observation should be discussed and put into the context with your findings? How do you exactly see these locuses being regulated through a continuum of cardiovascular disease - your results suggest more of an acute effect while this effects lessens and becomes nonsignificant as the time post-MI passes. How do you explain possible dynamics with respect to cavity sizes? This would definitely strengthen Discussion section where these aspects should be discussed.
  5. Why did you not use some sort of biomarker reflecting myocardial injury and myocyte mass damage such as troponin assay? In this way it would be possible to biochemically quantify potential differences in myocardial troponin leakage in wildtype vs. -/- animals. This should be discussed and if not feasible should be regarded as one of the limitations of the present study.

Author Response

We thank the reviewers for their questions, feedback, and insightful comments, which have greatly improved the submitted manuscript.

Reviewer 1 Comments and Suggestions for Authors

Authors present relevant preclinical data with respect to LINC00961/SPAAR locus loss on myocardial dynamics and cardiac response to myocardial infarction. They have previously identified that SPAAR is predominantly expressed in human cardiac endothelial cells and fibroblasts, while murine LINC00961 expression is hypoxia-responsive in mouse endothelial cells. Therefore, both locuses might have relevant implications for the development of cardiovascular disease. Their main result is that LINC00961-/- knockout mice showed a sex-specific delay in longitudinal growth and development, smaller left ventricular end- systolic and -diastolic areas and volumes, and greater risk area following myocardial infarction compared with wildtype littermates. Altogether, these data suggest a role for the LINC00961/SPAAR locus in cardiac endothelial cell and fibroblast cell function and hypoxic-response, as well as their development.

Ethical disclosures and animal protection protocols have been declared in the manscript and are well-described.

Methods are sufficiently described with no particular objections on my behalf.

I only have some minor comments:

  1. What would be possible clinical implications of your findings? I think it would be worthwhile to discuss possible outlook of your findings with respect to potential pharmacotherapeutics/treatment approaches in the future.

Addressed in lines 77-82, 192-199:

Thus, based on the association of LINC00961 with vascular endowment, the loss of LINC00961 could influence structure and function from prenatal life onward. From reduced vascular enrichment in the placenta and its influence on fetal and postnatal growth and development, to adult disease risk and pathological response particularly in the cardiovascular and pulmonary systems, LINC00961 loss of function could have clinical implications throughout the lifespan.

As we focused on the acute cardiac remodelling window, progression toward heart failure was outside of the scope of this current study. The apparent “normalization” of left ventricular volume and area in LINC00961-/- compared with wildtype mice at 7 days post-MI may suggest a relative acceleration in ventricular dilation in knockout mice. Although LINC00961-/- structural and functional cardiac deficiencies do not compromise homeostatic regulation at this young age, we hypothesise that disease severity and progression toward heart failure may be more pronounced during pathological remodelling. Future work investigating LINC00961 over-expression, for example using viral vector-based approaches for delivery to the heart, would elucidate the suitability of this lncRNA as a potential therapy.

  1. Were there any limitations to your study?

Addressed in lines 192-193:

As we focused on the acute cardiac remodelling window, progression toward heart failure was outside of the scope of this current study.

Addressed in lines 209-213:

Future work would benefit from assessment of sex-specific assessment of initial cardiac risk area, either directly or by proxy with a troponin assay. This study is limited by the CRISPR approach to global LINC00961 knockout, rather than with a conditional approach. Due to the influence of LINC00961 on vascular endowment and subsequent somatic growth and development, future work would benefit from a conditional knockout, either inducible or at the organs-specific level.

  1. How was myocardial area-at-risk estimated? Who did these observations and was there any inter-observer or intra-observer reliability issues? This should be explained at greater details.

Addressed in lines 295-298:

Briefly, stained hearts were cut into 1 mm transverse sections to the level above the suture, incubated in 2% TTC for 30 minutes at 37ºC, blotted dry and post-fixed in 4% formalin. Sections were scanned, and risk and infarct area quantified manually with Fiji by Dr Spiroski.

  1. In clinical practice, MIs are usually accompanied with an increase in end-systolic and end-diastolic volumes and pressures? This clinical observation should be discussed and put into the context with your findings? How do you exactly see these locuses being regulated through a continuum of cardiovascular disease - your results suggest more of an acute effect while this effects lessens and becomes nonsignificant as the time post-MI passes. How do you explain possible dynamics with respect to cavity sizes? This would definitely strengthen Discussion section where these aspects should be discussed.

As previously indicated, addressed in lines 192-199:

As we focused on the acute cardiac remodelling window, progression toward heart failure was outside of the scope of this current study. The apparent “normalization” of left ventricular volume and area in LINC00961-/- compared with wildtype mice at 7 days post-MI may suggest a relative acceleration in ventricular dilation in knockout mice. Although LINC00961-/- structural and functional cardiac deficiencies do not compromise homeostatic regulation at this young age, we hypothesise that disease severity and progression toward heart failure may be more pronounced during pathological remodelling. Future work investigating LINC00961 over-expression, for example using viral vector-based approaches for delivery to the heart, would elucidate the suitability of this lncRNA as a potential therapy.

  1. Why did you not use some sort of biomarker reflecting myocardial injury and myocyte mass damage such as troponin assay? In this way it would be possible to biochemically quantify potential differences in myocardial troponin leakage in wildtype vs. -/- animals. This should be discussed and if not feasible should be regarded as one of the limitations of the present study.

Discussed in Query 2, this is a limitation of the study. As we had approval through our University regulatory body to directly assess infarct size at in the non-recovery procedure, we did not pursue secondary assessment of 24-hour post-surgical infarct size by proxy with Troponin assay.

Reviewer 2 Report

In this research article, the authors have presented the role of LINC00961/SPAAR locus in murine growth and development, myocardial dynamics and cardiac response to MI. The authors have done a very good job explaining their research and the discussion section nicely summarized implications of the study and potential future studies. However, I have following suggestions on how to further improve this manuscript:

  1. The authors should elaborate more on the study design, in the present form its open to interpretation. A cartoon explaining the study design along with the proper time line would be an easy visual representation.

  1. Results are significantly underwritten.

  1. Statistical Analysis: Why was data log-transformed instead of using non-parametric statistical tests? Also, authors should explain how power analysis was done for example what was alpha value set for the analysis.

  1. Figure 1 A and 1 B should be explained, in the present form it doesn’t convey the intended message. How was UMAP analysis done, either add details to the “Materials or Methods” or explain in results. Additionally, use arrows to point to cardiac endothelial cells and fibroblasts clusters as its hard to figure it out from the legend.

  1. In figure 4 A, use arrows to point at MI risk area.

  1. Table1: Morphometric measurements should be normalized to tibial length (which stays constant throughout maturity) vs. body weight.

  1. Minor: incoherent sentences for example:

Line 22-23 “ To investigate the contribution of the  LINC00961/SPAAR locus to determination of longitudinal”

Line 29-31 “These data suggest a role for the LINC00961/SPAAR locus in cardiac endothelial cell and fibroblast cell function and hypoxic-response, and in growth and development, and basal cardiovascular function in adulthood.”

Line 25: SPAAR- expanded form where first used.

Author Response

We thank the reviewers for their questions, feedback, and insightful comments, which have greatly improved the submitted manuscript.

Reviewer 2 Comments and Suggestions for Authors

In this research article, the authors have presented the role of LINC00961/SPAAR locus in murine growth and development, myocardial dynamics and cardiac response to MI. The authors have done a very good job explaining their research and the discussion section nicely summarized implications of the study and potential future studies. However, I have following suggestions on how to further improve this manuscript:

  1. The authors should elaborate more on the study design, in the present form its open to interpretation. A cartoon explaining the study design along with the proper time line would be an easy visual representation.

Schematic of experimental animal generation and study period have now been included (Section 4.4, Figure 6)

  1. Results are significantly underwritten.

 Expanded/clarified as per Reviewers 1 and 3 suggestions

  1. Statistical Analysis: Why was data log-transformed instead of using non-parametric statistical tests? Also, authors should explain how power analysis was done for example what was alpha value set for the analysis.

Biological (physiological) variables in smaller experimental groups, despite being collected in sample sizes indicated by power calculations to produce statistical significance, may not result in normal distribution and variance of data. Data was log transformed to model proportional differences, which is generally more biologically relevant than using non-parametric statistical tests and reduces the probability of a type II error.

Power calculations included in section 4.8 (316-321)

  1. Figure 1 A and 1 B should be explained, in the present form it doesn’t convey the intended message. How was UMAP analysis done, either add details to the “Materials or Methods” or explain in results. Additionally, use arrows to point to cardiac endothelial cells and fibroblasts clusters as its hard to figure it out from the legend.

 Cell type indicated in key to clarify UMAP cell clustering. We appreciate the request to add arrows to indicate specific cell types, but we feel this would be repetitive after further clarification of cell type within the key. Heart Cell Atlas data visualisation clarified.

  1. In figure 4 A, use arrows to point at MI risk area.

 Arrows added

  1. Table1: Morphometric measurements should be normalized to tibial length (which stays constant throughout maturity) vs. body weight.

We appreciate the reviewer’s comments regarding growth and development. As this reviewer is aware, mouse strains and mutant lines vary widely in their longitudinal growth and development. We have characterised this line for the first time herein, and have determined appropriate parameters of longitudinal growth and development as required.

Although sexual maturity occurs at 4-5 weeks age, in relatively immature mice at 8-10 weeks of age adult body mass and size are not yet achieved and tibia length is not constant due to continued rapid growth until after 12 weeks of age. Due to the sex-specific differences in growth velocity of weight prior to the 8-week time point, organ weights were normalised to body weight which plateaus at 8-9 weeks age (Figure 2). Bone maturation was outside the scope of the current study and we are unable to confirm cessation of long bone growth at this early age.

  1. Minor: incoherent sentences for example:

Line 22-23 “ To investigate the contribution of the  LINC00961/SPAAR locus to determination of longitudinal”

This has been revised

Line 29-31 “These data suggest a role for the LINC00961/SPAAR locus in cardiac endothelial cell and fibroblast cell function and hypoxic-response, and in growth and development, and basal cardiovascular function in adulthood.”

Revised

Line 25: SPAAR- expanded form where first used.

First used in line 17, resolved

Reviewer 3 Report

In the manuscript "The influence of the LINC00961/SPAAR locus loss on murine development, myocardial dynamics, and cardiac response to myocardial infarction" the authors role discuss the role of LINC00961/SPAAR locus in cardiac endothelial cell, fibroblast cell function, and cardiovascular function.

Major comments:

None.

Minor comments:

I suggest the authors to better discuss clinical implications of their research in clinical practice. Improve methods presentation. Please, check text for typos and grammatical errors.

Author Response

We thank the reviewers for their questions, feedback, and insightful comments, which have greatly improved the submitted manuscript.

Reviewer 3 Comments and Suggestions for Authors

In the manuscript "The influence of the LINC00961/SPAAR locus loss on murine development, myocardial dynamics, and cardiac response to myocardial infarction" the authors role discuss the role of LINC00961/SPAAR locus in cardiac endothelial cell, fibroblast cell function, and cardiovascular function.

Major comments:

None.

Minor comments:

I suggest the authors to better discuss clinical implications of their research in clinical practice. Improve methods presentation. Please, check text for typos and grammatical errors.

Addressed in lines 77-82, 192-199:

Thus, based on the association of LINC00961 with vascular endowment, the loss of LINC00961 could influence structure and function from prenatal life onward. From reduced vascular enrichment in the placenta and its influence on fetal and postnatal growth and development, to adult disease risk and pathological response particularly in the cardiovascular and pulmonary systems, LINC00961 loss of function could have clinical implications throughout the lifespan.

As we focused on the acute cardiac remodelling window, progression toward heart failure was outside of the scope of this current study. The apparent “normalization” of left ventricular volume and area in LINC00961-/- compared with wildtype mice at 7 days post-MI may suggest a relative acceleration in ventricular dilation in knockout mice. Although LINC00961-/- structural and functional cardiac deficiencies do not compromise homeostatic regulation at this young age, we hypothesise that disease severity and progression toward heart failure may be more pronounced during pathological remodelling. Future work investigating LINC00961 over-expression, for example using viral vector-based approaches for delivery to the heart, would elucidate the suitability of this lncRNA as a potential therapy.

Round 2

Reviewer 2 Report

The Authors have done commendable job in addressing the concerns and improving the quality of Manuscript. I have no further concerns.

This manuscript is a resubmission of an earlier submission. The following is a list of the peer review reports and author responses from that submission.